# MobileNet-CA-YOLO: An Improved YOLOv7 Based on the MobileNetV3 and Attention Mechanism for Rice Pests and Diseases Detection

**Liangquan Jia** [1,†], **Tao Wang** [1,†], **Yi Chen** [2], **Ying Zang** [1], **Xiangge Li** [1], **Haojie Shi** [3] and **Lu Gao** [1,*]

1   School of Information Engineering, Huzhou University, Huzhou 313000, China; 02426@zjhu.edu.cn (L.J.); 2022388330@stu.zjhu.edu.cn (T.W.); 02750@zjhu.edu.cn (Y.Z.); 2021388222@stu.zjhu.edu.cn (X.L.)
2   School of Arts and Science, Fujian Medical University, Fuzhou 350122, China; chenyi0594@fjmu.edu.cn
3   College of Modern Agriculture, Zhejiang A&F University, Hangzhou 311300, China; shj@zafu.edu.cn
*   Correspondence: 02430@zjhu.edu.cn; Tel.: +86-183-6726-9616
†   These authors contributed equally to this work.

**Abstract:** The efficient identification of rice pests and diseases is crucial for preventing crop damage. To address the limitations of traditional manual detection methods and machine learning-based approaches, a new rice pest and disease recognition model based on an improved YOLOv7 algorithm has been developed. The model utilizes the lightweight network MobileNetV3 for feature extraction, reducing parameterization, and incorporates the coordinate attention mechanism (CA) and the SIoU loss function for enhanced accuracy. The model has been tested on a dataset of 3773 rice pest and disease images, achieving an accuracy of 92.3% and an mAP@.5 of 93.7%. The proposed MobileNet-CA-YOLO model is a high-performance and lightweight solution for rice pest and disease detection, providing accurate and timely results for farmers and researchers.

**Keywords:** MobileNetV3; rice pests and diseases; YOLOv7; coordinate attention mechanism; SIoU

## 1. Introduction

Rice is one of the main food sources in the world, with many countries, including China, relying on it as a staple food. By 2050, global rice production is expected to increase to about 650 million tons, which means a 40% increase in rice yield will be needed to meet the growing demand for food from a rising population [1]. However, increasing global food production faces some challenges. According to statistics, plant diseases and insect attacks worldwide cause annual crop yield losses ranging from 20% to 40%, resulting in global economic losses of USD 220 billion and USD 70 billion, respectively [2]. Traditional rice pest and disease recognition primarily rely on manual identification, which is inefficient and often leads to delayed detection. The widespread development of computer vision technology has been applied to crop pest and disease recognition in the agricultural field [3].

Classifier-based methods have been widely used in the identification of rice diseases and pests. These methods train models using machine learning algorithms, such as decision trees, support vector machines (SVMs) [4], and K-nearest neighbor (KNN) [5], to classify images based on their features. The advantages of these methods are their fast training speed and high accuracy. However, their disadvantage is that they require professional image processing techniques for complex diseases and pests, and their performance is limited by the quality and quantity of the training data [6]. Vakilian et al. [7] proposed an artificial neural network-based method for identifying beet armyworms, which achieved an average accuracy of 90%. YAO et al. [8] developed a handheld device for capturing rice pests and used algorithms, such as AdaBoost, for identifying and automatically counting rice planthoppers with a detection rate of 85.2%. In addition to the classifier-based methods, Goclawski et al. [9] and Zhou et al. [10] achieved good results in plant disease classification

using the unsupervised machine learning algorithm k-means. These machine learning-based technologies for agricultural pest detection require high-quality images, but in real-world field environments, the background of rice diseases and pests is complex and subject to weather conditions, making it difficult to obtain high-quality images. Manual methods are also ineffective in selecting color and shape features of rice pests from complex backgrounds, such as similar rice leaves and other non-target insects. Therefore, it is challenging to meet the demand for fully automated monitoring of rice pests using machine learning methods.

Deep learning has the advantage of automatically extracting features layer by layer and has its own feature generator, resulting in faster and more accurate recognition compared to machine learning [11]. This is particularly advantageous for detecting targets in complex backgrounds. TAN et al. [12] compared the recognition performance of deep learning and machine learning algorithms on the PlantVillage dataset [13] for identifying tomato leaf diseases. The results showed that deep learning algorithms had better precision, recall, and F1 values compared to machine learning algorithms. Meanwhile, Karar et al. [14] compared the accuracy of machine learning algorithms and CNN algorithms for pest detection on the IP102 dataset [15] and found that the detection accuracy of CNN algorithms was higher than machine learning algorithms. Sun et al. [16] incorporated an attention mechanism and developed a convolutional neural network model based on the attention mechanism to identify soybean aphids, achieving promising results. Overall, these studies demonstrate the superior performance of deep learning algorithms in object detection tasks compared to traditional machine learning algorithms.

Currently, research on the identification of rice pests and diseases is limited, with most studies focusing solely on detecting rice leaves. Additionally, existing models often have a high number of parameters, requiring powerful hardware and lacking practicality on mobile devices. To address these limitations, this article introduces a novel approach for rice pest and disease detection based on an improved YOLOv7 algorithm. The proposed method utilizes YOLOv7 as the object detection framework and MobileNetV3 as the backbone network. To fully extract spikelet features, the CA (coordinate attention) mechanism is incorporated into the feature fusion part of YOLOv7. Moreover, the SIoU loss function is employed to enhance convergence speed. By optimizing the YOLO algorithm, the proposed method achieves efficient recognition of multiple objects with lower power consumption, which is advantageous for mobile device applications. The contributions of this article can be summarized as follows:

(1) To enable better integration with mobile devices, this experiment used different lightweight networks to replace the YOLOv7 backbone network, reducing the number of parameters and increasing the speed of the model, resulting in the most suitable network structure for rice pest and disease detection.

(2) The coordinate attention (CA) module was used to focus on the channel number of the feature map, enhancing the representation of the target and further improving the accuracy of rice pest and disease recognition.

(3) To improve the model's generalization ability and accelerate training convergence speed, this experiment compared different loss functions and ultimately selected SIoU as the bounding box loss function.

(4) To ensure the model achieves the best performance in rice pest and disease recognition, the "rice pest and disease" object detection dataset was created in our laboratory.

## 2. Materials and Methods

### 2.1. Dataset

Crop pests and diseases pose a serious threat to agriculture [17], and traditional monitoring and diagnosis methods are time-consuming and prone to misdiagnosis. Deep learning technology, particularly computer vision and image recognition algorithms, has become a hot research topic for crop pest and disease identification [18]. However, training deep learning models requires a large amount of data, making image datasets an important

component. By collecting, annotating, and organizing large-scale crop pest and disease image datasets, deep learning models can be trained with sufficient data and samples, improving their accuracy and generalization ability. The establishment of these datasets also helps accelerate the diagnosis and control of pests and diseases, providing effective technical support for the agricultural industry.

In this experiment, most of the image data was obtained from the Baidu image library. We collected three types of disease images and three types of pest images through internet channels, which included rice blast, rice false smut, bacterial leaf blight, rice borer, rice planthopper, and rice locust image data. The image data is shown in Figure 1.

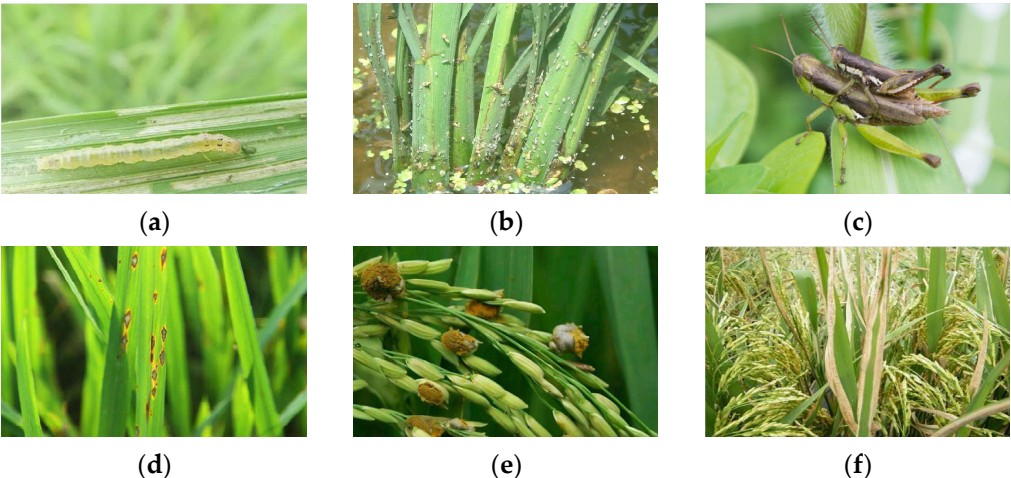

**Figure 1.** Image samples in the dataset. (**a**) Rice borer; (**b**) rice planthopper; (**c**) rice locust; (**d**) rice blast; (**e**) rice false smut; (**f**) bacterial leaf blight.

To ensure the diversity and quantity of the collected data, we utilized web crawlers to collect the images from various sources. In addition, we selected images that contained a clear and complete target through manual selection to ensure the quality of the data.

The rice plant disease and pest datasets, which are mainly used to train deep learning models in our experiment, are challenging to obtain. The datasets are often limited and expensive to collect and annotate. Therefore, we utilized a combination of internet-based data collection methods and manual selection to compile a sufficient and diverse dataset to improve the accuracy and robustness of our model.

In order to improve the quality and efficiency of data annotation, this study adopted the Makesense AI platform for data annotation and processing. Makesense AI is an open-source online annotation tool that can be directly operated on a webpage without the need for download or installation. The platform supports multiple annotation types, such as points, lines, and boxes, and also supports various label export formats, including YOLO, VOC XML, VGG JSON, and CSV, among others. Additionally, Makesense AI also provides a pre-trained object detection model, SSD, to assist in manual annotation and significantly improve annotation efficiency and accuracy.

During the annotation process using the Makesense AI platform, the research team established strict annotation standards and selected professional annotators from within the team. Annotators were able to quickly and accurately annotate large amounts of rice image data with Makesense AI's image annotation tool while ensuring the quality of annotations.

### 2.1.1. Data Augmentation

Rice plant disease and pest datasets are characterized by their diversity, rotational invariance, and symmetry, making them suitable for data augmentation [19]. Previous studies have used online data augmentation to enhance the diversity of training images and improve the generalization capability of the network for datasets with sufficient data samples but uneven image distribution and single sample types. Online data augmentation

involves applying image enhancement strategies to each image batch that inputs into the network. Specifically, various function mappings are applied to each image in the batch, which generates a series of enhanced images to be trained and then put into the neural network. This image augmentation method is random in nature and can enhance the model's generalization ability. However, as this study was based on a small-scale image dataset collected by the research team, we chose offline data augmentation to prevent overfitting and low model generalization during model training. By applying different forms of function mappings to limited data and storing the transformed data in a local dataset, the dataset can be expanded. Content and geometric transformations were used in our experiment to cater to rice plant disease and pest characteristics for offline data augmentation. Geometric transformations were adopted to modify image attributes without altering the image's content, such as image cropping (Random Resize Crop, RPC), image translation, and image rotation [20]. Content transformation includes color jittering, adding noise, and so on. The combination of geometric transformations and content transformations allows for a comprehensive modification of image attributes, enhancing the robustness and generalization capability of the model. The specific process is illustrated in Figure 2. In Table 1, we present the classification and corresponding quantities of images in the enhanced dataset for rice pests and diseases.

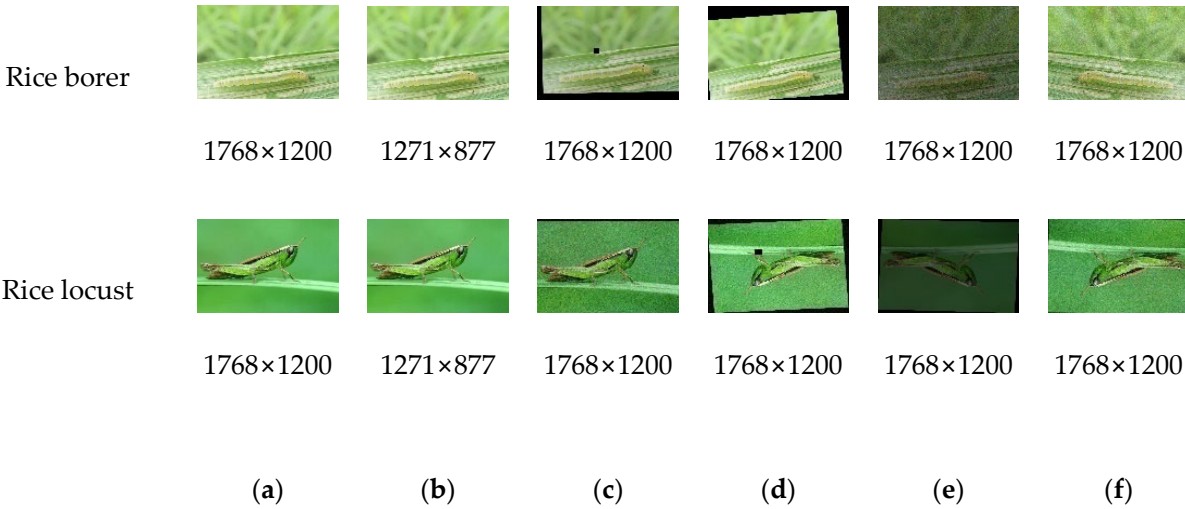

**Figure 2.** Image samples in the data augmentation. (**a**) Original image; (**b**) resize; (**c**) translation; (**d**) rotation; (**e**) contrast ratio; (**f**) noising.

**Table 1.** Dataset information.

| Classification | Number of Images |
|---|---|
| Rice borer | 630 |
| Rice planthopper | 634 |
| Rice locust | 630 |
| Rice blast | 622 |
| Rice false smut | 627 |
| Bacterial leaf blight | 630 |

### 2.1.2. Dataset Partitioning

In this study, we used the rice plant disease and pest dataset collected by our team as the training and testing datasets for object detection. The dataset was randomly split into a training set, a validation set, and a test set in an 8:1:1 ratio. The training set was utilized to train the model parameters, the validation set was used to adjust hyperparameters and prevent overfitting, and the test set was employed to evaluate the model's performance [21]. To ensure an even distribution of samples within the split datasets, we randomly shuffled

the order of the samples in the dataset. We analyzed the class and image size distributions in the dataset after the shuffle, and then adjusted the split ratio to meet the experimental requirements and obtain better model performance.

### 2.2. MobileNet-CA-YOLO

### 2.2.1. MobileNetV3 Module

MobileNetV3 [22] is a deep neural network architecture that was obtained through a network architecture search (NAS). It inherits the depthwise separable convolution from MobileNetV1 and the linear bottleneck residual structure from MobileNetV2 [23]. The most impressive improvement of MobileNetV3 is the addition of the Squeeze-and-Excitation (SE) structure to the bottlenet structure, as well as the substitution of swish with h-swish. The swish and h-swish formulas are shown below. Due to the long computation time of sigmoid, especially on mobile devices, h-swish was used to approximate sigmoid. ReLU has several benefits, including the ability to perform calculations on any hardware or software platform, eliminating potential accuracy loss during quantization, and being more pronounced in deep networks. MobileNetV3 is designed for efficient image classification and object detection tasks in computer vision. It is designed to be efficient, meaning it requires fewer computing resources compared to other architectures while still maintaining high precision. The switch formula is shown in Equation (1), and the h-swish formula is shown in Equation (2). The main network structure of MobileNetV3 is shown in Figure 3, with detailed parameter information shown in Table 2.

$$\text{swish } x = x \cdot \delta(x), \tag{1}$$

$$\text{h-swish} = x \cdot [\text{ReLU6}(x + 3)/6], \tag{2}$$

Equation (1) represents the swish activation function, which applies the element-wise multiplication of the input "x" with the sigmoid function $\delta(x)$. It amplifies positive values and diminishes negative values, introducing non-linearity to the computation.

Equation (2) represents the h-swish activation function, a more computationally efficient version of swish. It applies the element-wise multiplication of the input "x" with the result of the ReLU6 function applied to the sum of "x" and 3, divided by 6.

For our experiment, we have selected MobileNetV3-small as the backbone network for feature extraction in YOLOv7. By using MobileNetV3-small, which is specifically designed for resource-constrained environments, we aim to efficiently extract important features from the input data.

In this approach, we utilize the feature extraction layer of MobileNetV3-small, which includes the layers before the pooling layer mentioned in the table. By focusing on the feature extraction layer, we can effectively capture relevant and discriminative features from the input data.

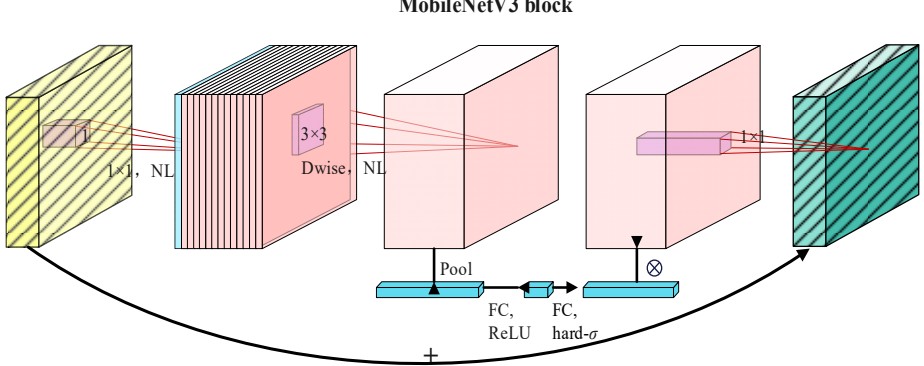

**Figure 3.** MobileNetV3 principal network architecture.

**Table 2.** MobileNet V3 network parameter information.

| Input | Operator | Exp Size | #out | SE | NL | S |
|---|---|---|---|---|---|---|
| $224^2 \times 3$ | conv2d, $3 \times 3$ | - | 16 | - | HS | 2 |
| $112^2 \times 16$ | bneck, $3 \times 3$ | 16 | 16 | $\checkmark$ | RE | 2 |
| $56^2 \times 16$ | bneck, $3 \times 3$ | 72 | 24 | - | RE | 2 |
| $28^2 \times 24$ | bneck, $3 \times 3$ | 88 | 24 | - | RE | 1 |
| $28^2 \times 24$ | bneck, $5 \times 5$ | 96 | 40 | $\checkmark$ | HS | 2 |
| $14^2 \times 40$ | bneck, $5 \times 5$ | 240 | 40 | $\checkmark$ | HS | 1 |
| $14^2 \times 40$ | bneck, $5 \times 5$ | 240 | 40 | $\checkmark$ | HS | 1 |
| $14^2 \times 40$ | bneck, $5 \times 5$ | 120 | 48 | $\checkmark$ | HS | 1 |
| $14^2 \times 48$ | bneck, $5 \times 5$ | 144 | 48 | $\checkmark$ | HS | 1 |
| $14^2 \times 48$ | bneck, $5 \times 5$ | 288 | 96 | $\checkmark$ | HS | 2 |
| $7^2 \times 96$ | bneck, $5 \times 5$ | 576 | 96 | $\checkmark$ | HS | 1 |
| $7^2 \times 96$ | bneck, $5 \times 5$ | 576 | 96 | $\checkmark$ | HS | 1 |
| $7^2 \times 96$ | conv2d, $1 \times 1$ | - | 576 | $\checkmark$ | HS | 1 |
| $7^2 \times 576$ | pool, $7 \times 7$ | - | - | - | - | 1 |
| $1^2 \times 576$ | conv2d $1 \times 1$, NBN | - | 1024 | - | HS | 1 |
| $1^2 \times 1024$ | conv2d $1 \times 1$, NBN | - | k | - | - | 1 |

This approach aligns with our experimental design, where the primary objective is to detect and identify rice pests and diseases in real-world scenarios. By leveraging the efficient architecture of MobileNetV3-small, we can extract essential features that are crucial for accurate detection.

By utilizing the feature extraction layer of MobileNetV3-small, we can benefit from its inherent ability to capture rich representations without the need for additional training or transfer learning. This ensures that our model is specifically tailored to the task of rice pest and disease detection.

### 2.2.2. Add the CA Attention Module

Adding attention mechanisms to neural networks is an efficient method for improving their feature extraction capabilities. One such mechanism is the coordinate attention (CA) module proposed by Hou et al. [24]. This module is highly flexible and can be easily integrated with various classic network structures to improve the feature extraction capabilities of the network.

The CA module relies on channel information and divides attention into two 1D feature encodings, which aggregate features separately along two independent spatial directions. Integrating the CA attention mechanism into the model can enhance the model's perception of different spatial positions and targets. This method not only captures dependency relationships along one spatial direction but also preserves accurate position information along the other spatial direction [25]. The introduction of the coordinate attention mechanism enables the model to focus on more semantic information [26], thus improving its generalization ability. The structure of the CA module is illustrated in Figure 4.

Incorporating the coordinate attention mechanism (CA) can provide more semantic information for the YOLOv7 object detection model, thereby improving its detection accuracy and efficiency. Specifically, the CA can assist the model in more accurately locating regions of interest and identifying properties, such as the shape and size of the detected objects [27]. In addition, introducing the CA can reduce unnecessary computational complexity, thereby improving the efficiency of the detection algorithm. Therefore, the attention mechanism is an effective means of optimizing the YOLOv7 model.

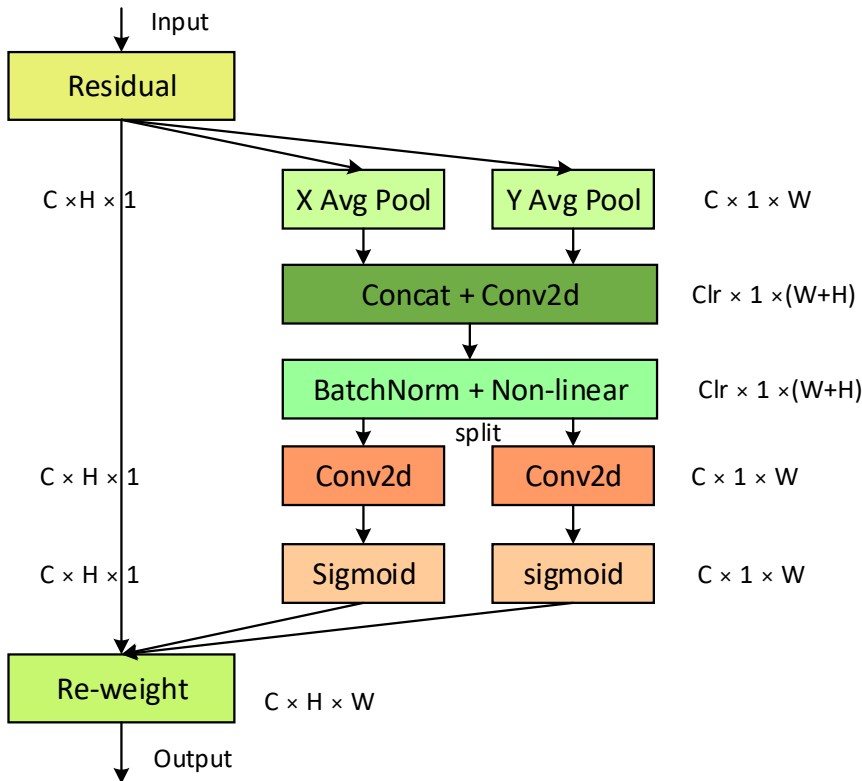

**Figure 4.** The structure of the CA attention module.

### 2.2.3. SIoU Loss Function

Object detection is one of the core problems in computer vision, and its effectiveness largely depends on the loss function used [28]. Traditional object detection loss functions are typically based on the aggregation of bounding box regression metrics, such as the distance, overlapping area, and aspect ratio (e.g., GIoU, CIoU, ICIoU, etc.) between the predicted and ground truth boxes [29]. However, existing methods often overlook the problem of direction mismatch between the ground truth and predicted boxes, which leads to drifting of the predicted boxes during the training process, resulting in slow convergence and poor performance of the model. To address this issue, Gevorgyan [30] proposed an SIoU loss function. This method redefines the loss function in object detection tasks, which consists of four parts: angle loss, distance loss, shape loss, and IoU loss. Specifically, the angle loss considers the angle between the vectors of the ground truth and predicted boxes, the distance loss considers the distance between the ground truth and predicted boxes, the shape loss considers the shape difference between the ground truth and predicted boxes, and the IoU loss calculates the intersection over union between the predicted and ground truth boxes to evaluate the accuracy of the predicted boxes. By introducing these four loss functions, the accuracy and robustness of the object detection model can be improved while avoiding overfitting issues. The calculation of angle contribution cost in the SIoU loss function is shown in Figure 5, and the final definition of the SIoU loss function is shown in Equation (3).

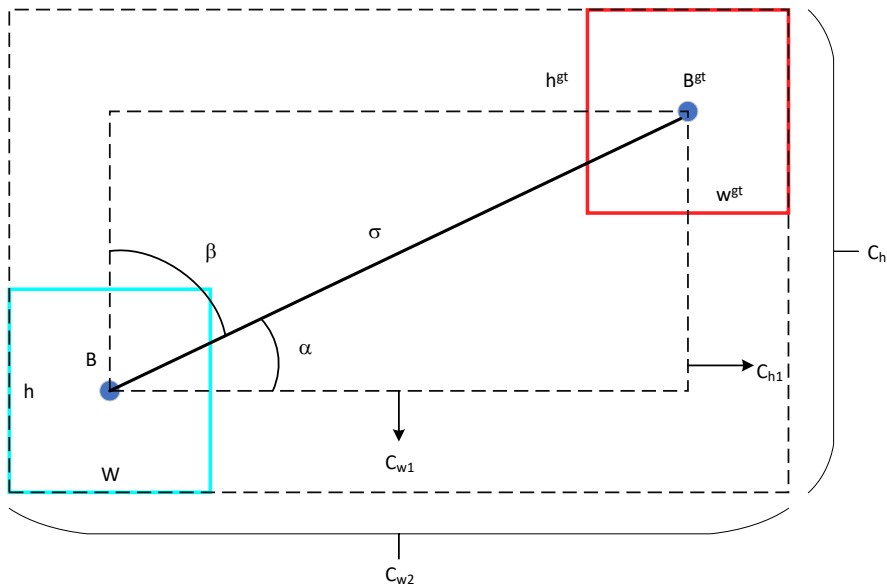

**Figure 5.** Predicted and ground truth box vector angles.

$$
\begin{cases}
\Lambda = 1 - 2 * sin^2\left(\arcsin\left(\frac{c_{h1}}{\sigma}\right) - \frac{\pi}{4}\right) \\[2mm]
\sigma = \sqrt{\left(b_{c_x}^{gt} - b_{c_x}\right)^2 + \left(b_{cy}^{gt} - b_{c_y}\right)^2} \\[2mm]
\Delta = \sum_{t=x,y}\left(1 - e^{-\gamma\rho_t}\right) = 2 - e^{-\gamma\rho_x} - e^{-\gamma\rho_y} \\[2mm]
\rho_x = \left(\frac{b_{c_x}^{gt} - b_{c_x}}{c_{w2}}\right)^2, \rho_y = \left(\frac{b_{cy}^{gt} - b_{c_y}}{c_{h2}}\right)^2, \gamma = 2 - \Lambda \\[2mm]
\Omega = \sum_{t=w,h}\left(1 - e^{-\omega_t}\right)^\theta \\[2mm]
\omega_w = \frac{|w - w^{gt}|}{max(w, w^{gt})}, \omega_h = \frac{|h - h^{gt}|}{max(h, h^{gt})} \\[2mm]
Loss_{IOU} = 1 - IoU + \frac{\Delta + \Omega}{2} \\[2mm]
IOU = \frac{|B \cap B^{gt}|}{|B \cup B^{gt}|}
\end{cases}
\tag{3}
$$

In Equation (3), $\Lambda$ is the angle loss function, where $\sigma$ represents the center point distance between the predicted box and the ground truth box, $b_{c_x}^{gt}$ and $b_{cy}^{gt}$ are the center point coordinates of the ground truth box, and $b_{c_x}$ and $b_{c_y}$ are the center point coordinates of the predicted box. $\Delta$ is the distance loss function, in which $C_{w2}$ and $C_{h2}$ represent the width and height of the bounding box that encompasses both the ground truth and predicted boxes. $\Omega$ denotes the shape loss function, where $w^{gt}$, $h^{gt}$, $w$, and $h$ denote the width and height of the ground truth and predicted boxes, respectively.

### 2.2.4. Improved YOLOv7 Network Architecture

YOLO is a fast, accurate, and high-quality one-stage object detection algorithm that is often chosen as the preferred option for object detection. Based on deep learning, YOLOv7 performs object detection on images by simultaneously detecting multiple objects and providing their precise bounding boxes in a single forward pass. YOLOv7 outperforms most detectors in terms of speed and accuracy and has become the preferred model for many real-time applications [31].

Combining YOLOv7 with MobileNetV3 enables the advantages of both algorithms to be fully utilized, achieving fast and accurate object detection. YOLOv7 provides an efficient object detection algorithm, while MobileNetV3 has efficient computational speed and low memory consumption, making the combined model suitable for real-time applications

and enabling deep models to be better ported to embedded devices. By introducing a coordinate attention mechanism (CA) to the YOLOv7 object detection model based on the MobileNetV3 architecture, more semantic information can be provided, thereby improving the precision and efficiency of detection. The network architecture of MobileNet-CA-YOLO is shown in Figure 6.

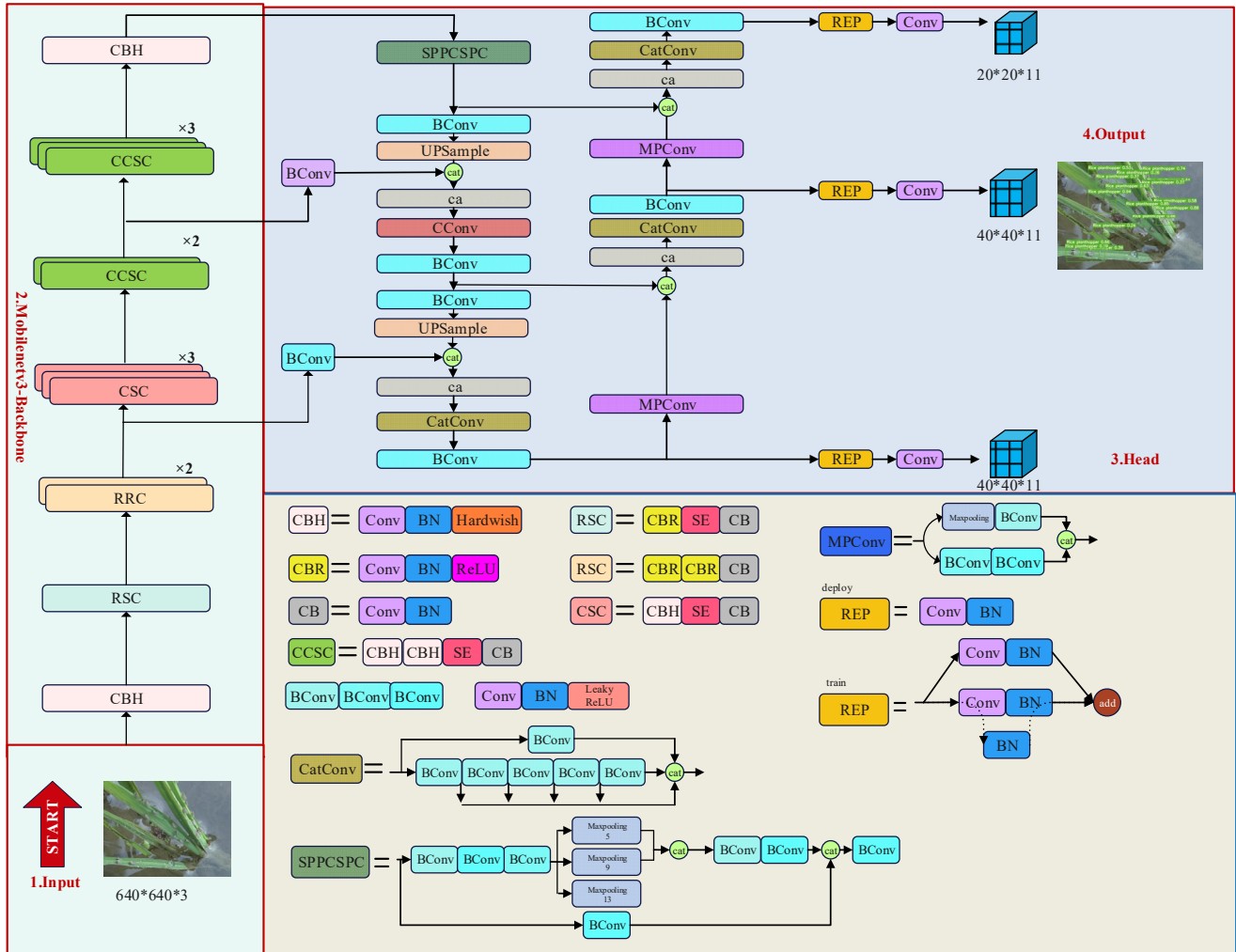

**Figure 6.** MobileNet-CA-YOLO network structure.

### 2.3. Model Evaluation Criteria

In order to evaluate the effectiveness of the MobileNet-CA-YOLO network model in rice disease and pest detection, this study selects precision (P), recall (R), F1 score, and mean average precision (mAP) as evaluation metrics [32]. In the mAP evaluation metric, the experiment uses mAP@.5 with an IoU threshold of 0.5 as the evaluation metric. The calculation formula for precision is shown in Equation (4), the formula for recall is shown in Equation (5), the formula for F1 score is shown in Equation (6), and the formula for mAP is shown in Equation (7).

$$\text{Precision} = \frac{\text{TP}}{\text{TP} + \text{FP}} \tag{4}$$

$$\text{recall} = \frac{\text{TP}}{\text{TP} + \text{FN}} \tag{5}$$

$$\text{F1} = \frac{2\text{TP}}{2\text{TP} + \text{FN} + \text{FP}} \tag{6}$$

$$mAP = \frac{\sum AP}{n} \tag{7}$$

In Equations (4)–(7), TP represents the number of regions of rice disease and pests correctly detected, FP represents the number of regions of rice disease and pests that were not correctly detected, and FN represents the number of regions of rice disease and pests incorrectly detected. The AP value is the area under the curve formed by the combination of different precision (P) and recall (R) points, with n representing the number of classes.

## 3. Results and Analysis

### 3.1. Experimental Platform and Parameter Settings

The model training was conducted on a Linux operating system using the PyTorch training and testing framework. The server was equipped with an Intel(R) Xeon(R) CPU E5-2650 v3 @ 2.30 GHz processor, 64 GB of RAM, and an NVIDIA TITAN Xp 12 GB graphics card, and was installed with the CUDA 11.2 parallel computing framework and the CUDNN 8.2 deep neural network acceleration library. The input image size was set to 640 pixels × 640 pixels, the batch size was set to 12, the training steps were set to 200, the learning rate was 0.01, the momentum was set to 0.937, and random stochastic gradient descent (SGD) optimization with a weight decay of 0.005 was used. The model was trained on the server to save time and was subsequently validated locally, with detailed environment configurations provided in Table 3. The training parameter settings are shown in Table 4.

**Table 3.** Environment configuration.

| Configuration | | Local Configuration | Server Configuration |
|---|---|---|---|
| Hardware | CPU | Inter Core i3 12100 | Inter Core E5-2650 |
| | GPU | GeForce GTX 3060-12G | GeForce RTX 3080ti-12G |
| | RAM | 16 GB | 64 GB |
| Software | System | Win 11 | Ubuntu 22.04 |
| | Python | 3.8 | 3.8 |
| Environment | Pytorch | 1.8 | 1.8 |
| | CUDA | 11.2 | 11.2 |
| | Cudnn | 8.2 | 8.2 |

**Table 4.** Parameter Settings.

| Parameter | Values |
|---|---|
| Base learning rate | 0.01 |
| End learning rate | 0.1 |
| Momentum | 0.937 |
| Batch size | 12 |
| Learning rate policy | SGD |
| Droupout | 0.0005 |
| Epoch | 200 |

### 3.2. Evaluation of the MobileNet-CA-YOLO Experimental Results

#### 3.2.1. Comparison of the Results of Different Models

In order to demonstrate the superior application of the MobileNet-CA-YOLO network on mobile devices, with higher computational efficiency and better performance in rice disease and pest detection, we conducted comparative experiments on parameters, GPU consumption, and mAP values between the original YOLOv7 and YOLOv7-MobileNetV3, MobileNet-CA-YOLO, YOLOv7-tiny, YOLOv7-shuffleNetV2-CA, and YOLOv-MobileNetV2-CA. The results are shown in Table 5, and the model performance comparison is presented in Figure 7.

**Table 5.** Analysis of the experimental effects of the different models.

| Model | Parameters | GPU/% | mAP@.5/% | mAP@.5:.95/% | FPS |
|---|---|---|---|---|---|
| YOLOv7 | 37.223M | 72.3 | 94.6 | 78.14 | 52.08 |
| YOLOv7-MobileNet V3 | 6.596M | 23.25 | 91.3 | 63.2 | 71.42 |
| YOLOv7-shuffleNetV2-CA | 6.999M | 22 | 86.5 | 56.5 | 85.47 |
| YOLOv7-MobileNetV2-CA | 6.604M | 32.5 | 90.8 | 63 | 78.13 |
| MobileNet-CA-YOLO | 6.956M | 28 | 93.7 | 67 | 84.74 |
| YOLOv7-tiny | 6.2M | 20.8 | 88 | 51.5 | 87.7 |

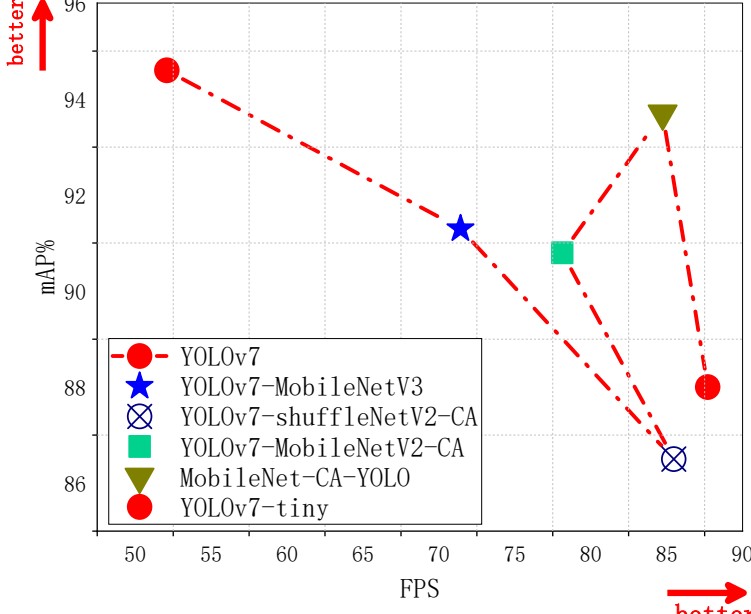

**Figure 7.** Compared with other models, the model in this experiment can achieve better performance in detecting rice pests and diseases.

Based on the analysis results in Table 5, our proposed MobileNet-CA-YOLO model exhibits a significant reduction in parameter size, accounting for only 18.69% of the original YOLOv7 model. Compared to the YOLOv7-MobileNetV3 model with MobileNetV3 as the backbone network, the MobileNet-CA-YOLO model only increases the parameter size by 0.36M. When considering the same batch size setting, the GPU memory consumption of the MobileNet-CA-YOLO model is merely 28%, while the YOLOv7 model consumes 72.3% of GPU memory. In terms of mAP evaluation, the MobileNet-CA-YOLO model outperforms the YOLOv7-MobileNetV3 model by 2.4 percentage points and surpasses the YOLOv7-tiny model by 5.7 percentage points. Additionally, regarding the FPS evaluation metric, the MobileNet-CA-YOLO model demonstrates a 32.66 FPS improvement over the YOLOv7 model and a 13.45 FPS improvement over the YOLOv7-MobileNetV3 model, with only a marginal 2.96 FPS reduction compared to the YOLOv7-tiny model. Figure 7 illustrates the superior performance of our proposed model in rice disease and pest detection. It is worth noting that the YOLOv7-tiny model achieves the fastest recognition speed but at the expense of lower accuracy. On the other hand, the YOLOv7 model exhibits the highest recognition accuracy but lags behind in terms of recognition speed compared to the other models. These findings highlight that the MobileNet-CA-YOLO model not only advances parameter reduction but also significantly improves recognition speed and reduces hardware requirements, making it more suitable for rice disease and pest detection.

3.2.2. Comparison of the Results of Different IoU Loss Functions

In this experiment of training the MobileNet-CA-YOLO network for rice disease and pest detection, we compared the SIoU, CIoU, and WIoU loss functions, and the results are

shown in Table 6. The comparison of precision and recall under different loss functions is shown in Figure 8.

**Table 6.** Comparison of the recognition effects of the different loss functions.

| IoU Loss Function | P/% | mAP@.5/% | mAP@.5:.95/% | FPS |
|---|---|---|---|---|
| CIoU | 89.7 | 91.4 | 65 | 75.1 |
| WIoU | 91.6 | 91.7 | 63.2 | 76.9 |
| SIoU | 92.3 | 93.7 | 67 | 84.74 |

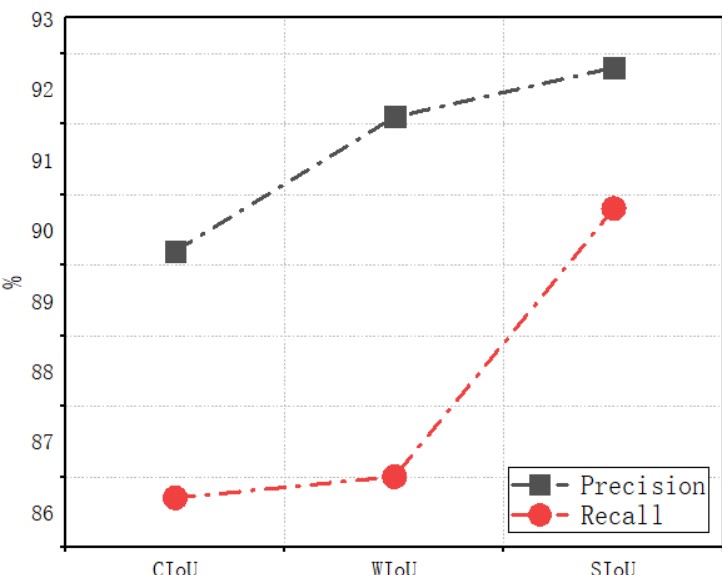

**Figure 8.** Comparison of precision and recall under the different loss functions.

According to the analysis in Table 6, the MobileNet-CA-YOLO model using the SIoU loss function achieves the highest precision, which is 1.3 percentage points higher than the WIoU loss function and 2.6 percentage points higher than the default CIoU loss function. In addition, the MobileNet-CA-YOLO model using the SIoU loss function is also faster than the other two loss functions. As shown in Figure 8, the model using the SIoU loss function performs better in both precision and recall than the other two loss functions. Therefore, we believe that selecting the SIoU loss function as the bounding box loss function for the MobileNet-CA-YOLO model is the best choice.

3.2.3. MobileNet-CA-YOLO Experimental Results Analysis

The confusion matrix is a vital tool in machine learning and is used to compare classification results with actual predictions. It allows us to display each classification result of the model training process clearly and conveniently. In this experiment, the confusion matrix is mainly used to evaluate the performance of the MobileNet-CA-YOLO detection algorithm. It is a two-dimensional table-shaped matrix where rows represent the actual categories and columns represent the predicted categories. By counting the prediction results between different categories, various metrics such as accuracy, recall, and misidentification rate can be determined. The confusion matrix is shown in Figure 9, and the recognition results of each category are shown in Table 7.

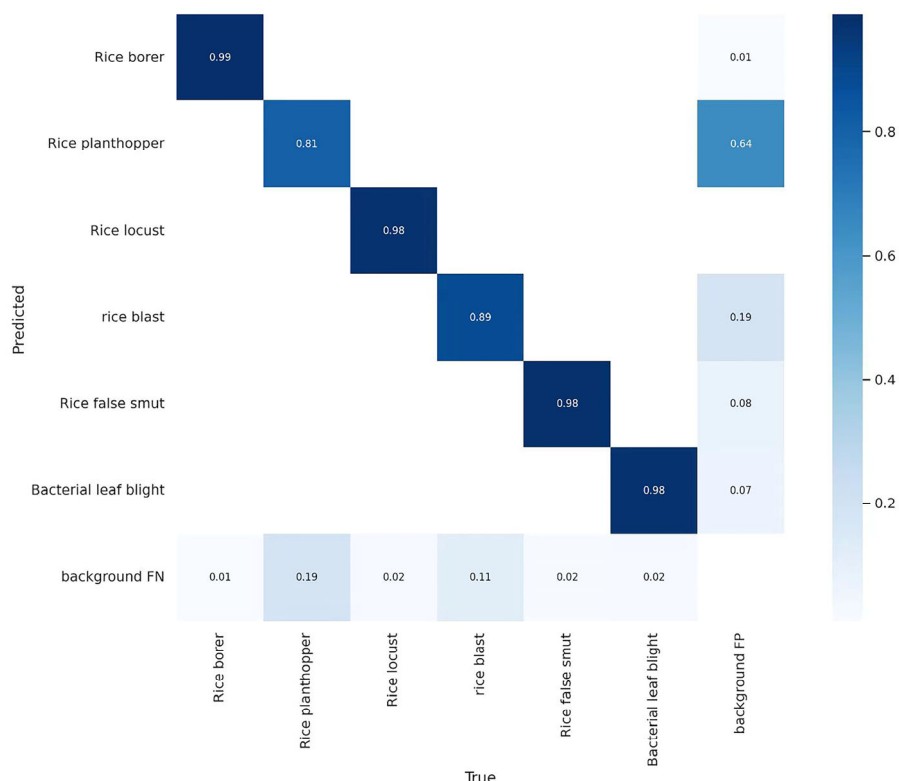

**Figure 9.** Confusion matrix.

**Table 7.** Identification effect of various rice diseases and pests.

| Classification | P/% | mAP@.5/% | mAP@.5:.95/% |
|---|---|---|---|
| Rice borer | 95.8 | 97.8 | 73.4 |
| Rice planthopper | 78.2 | 78.8 | 42.5 |
| Rice locust | 98.3 | 99.1 | 83.1 |
| Rice blast | 86.9 | 89.5 | 56 |
| Rice false smut | 93.7 | 98.4 | 71.3 |
| Bacterial leaf blight | 93.1 | 98 | 75.4 |

Based on the analysis of the confusion matrix, in this experiment, rice diseases and pests were divided into six categories, including three pests: rice borer, rice planthopper, and rice locust, and three diseases: rice blast, rice false smut, and bacterial leaf blight. Taking rice blast as an example, it can be observed that the model achieved a classification accuracy of 89% for rice blast. The false negative rate, which represents the rate at which rice blast was incorrectly predicted as the background, was 11%. The false positive rate, indicating the rate at which the background was mistakenly identified as rice blast, was 19%. Analyzing Table 6, it can be seen that the model performed best in identifying rice borer and rice locust, with accuracies of 95.8% and 98.3%, respectively. However, the recognition accuracy for rice planthopper was slightly lower at 78.2%.

In this experiment of training rice diseases and pests using the MobileNet-CA-YOLO network, the F1_curve, PR_curve, P_curve, and R_curve during the model training process are shown in Figure 10.

These curves provide insights into the model's learning performance for the six different types of rice diseases and pests. The F1_curve reflects the accuracy of the model's identification of different rice diseases and pests. The PR_curve can clearly show the mAP@.5 value for different rice diseases and pests, where the larger the area enclosed by the curve on the *x*-axis and *y*-axis, the better the performance.

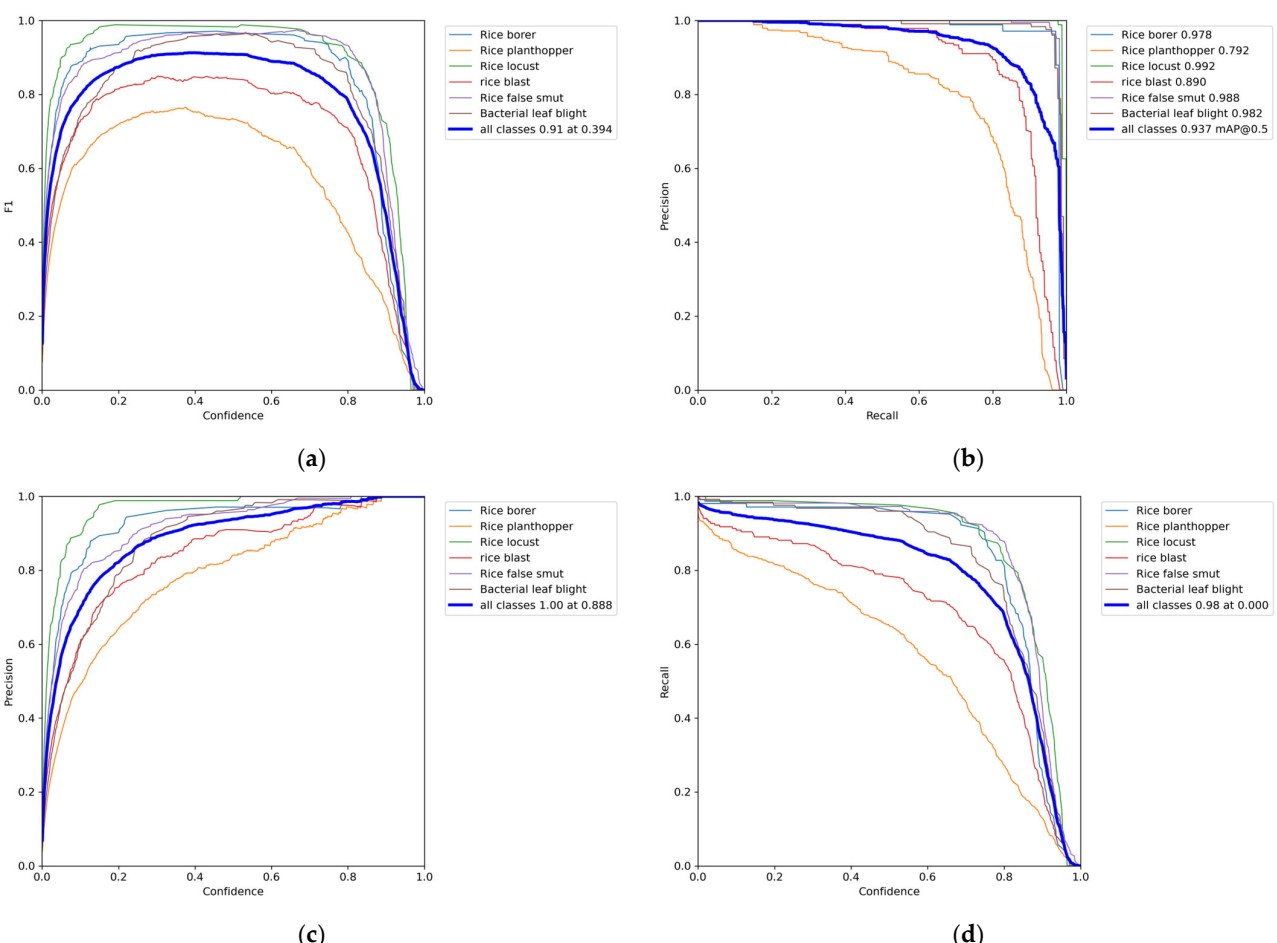

**Figure 10.** (**a**) F1_curve; (**b**) PR_curve; (**c**) P_curve; (**d**) R_curve.

In this experiment, a batch size of 12 was used, so 12 images were taken for training each time. The YOLO algorithm provides good visualization effects. After training starts, train*.jpg images can be viewed to observe the training images, labels, and augmentation effects. After each epoch of training, test_batch*_gt.jpg can be used to see the ground truth bounding boxes on the validation set, and test_batch*_pred.jpg can be used to see the predicted bounding boxes for each epoch. An example of the predicted results on the validation set during model training is shown in Figure 11.

Based on the analysis in Figure 11, it can be concluded that the model performed well on the validation set after multiple epochs of training. Specifically, the model learned more easily when the target object was larger and there were fewer targets in the image, such as rice borer and rice locust. However, it was more difficult for the model to learn in situations where the target object was smaller and there were too many targets in the image, such as rice planthopper, rice blast, and bacterial leaf blight, especially for diseases with few and similar features, such as rice blast and bacterial leaf blight. Through multiple iterations of training, the MobileNet-CA-YOLO model demonstrated significant improvements, achieving impressive accuracy metrics. Specifically, the model achieved an F1 score of 91%, a precision of 92.3%, and an mAP@.5 of 93.7%. It is worth noting that the model's performance varied across different classes, with the lowest precision observed in the rice planthopper class at 78.2% and the rice blast class at 86.9%. This comprehensive evaluation underscores the model's effectiveness while acknowledging the variability in accuracy rates among different categories.

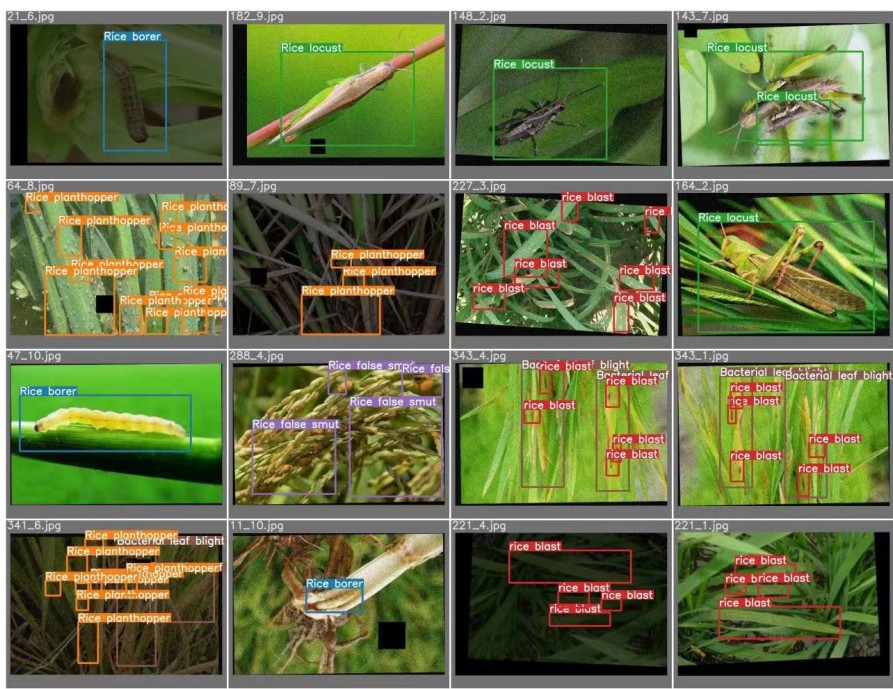

**Figure 11.** Validation set prediction results.

### 3.2.4. Analysis of MobileNet-CA-YOLO Test Results

During the inference testing process, the highest accuracy model saved from the training process was used for testing. In this experiment, three models, YOLOv7, YOLOv7-MobileNet V3, and MobileNet-CA-YOLO, were used to identify rice diseases and pests in the test images. The recognition results are shown in Figure 12.

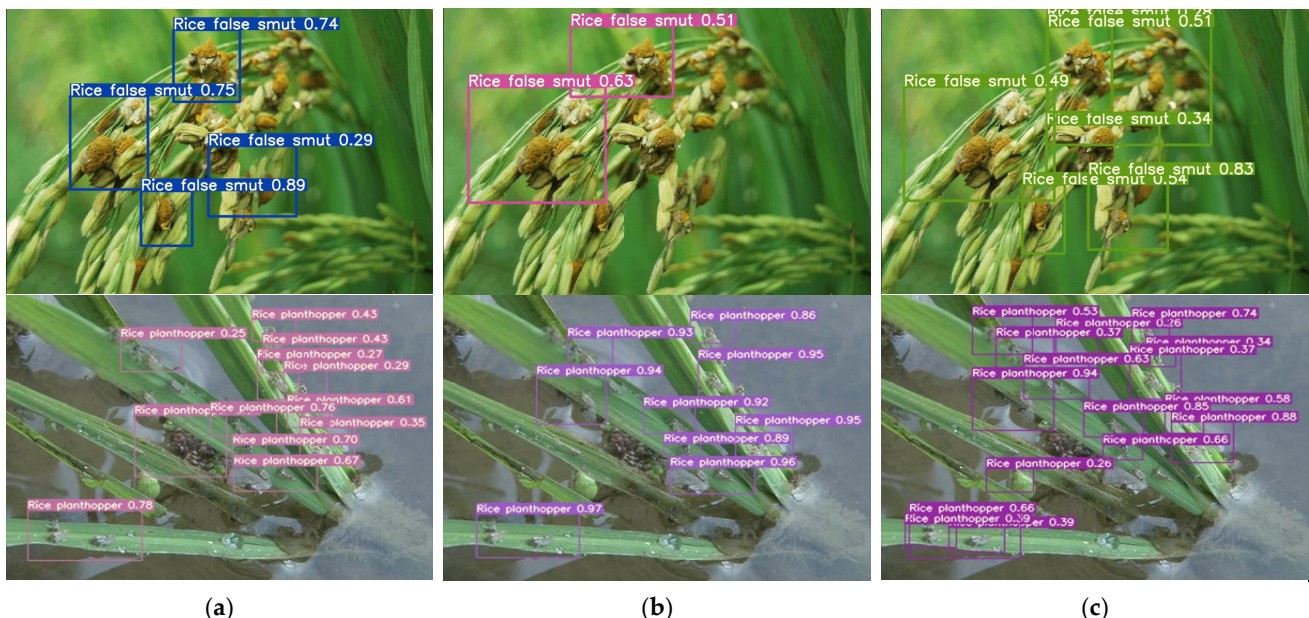

**Figure 12.** Recognition result. (**a**) YOLOv7; (**b**) YOLOv7-MobileNet V3; (**c**) MobileNet-CA-YOLO.

The comparison results of the YOLOv7, YOLOv7-MobileNet V3, and MobileNet-CA-YOLO models are shown in Figure 12 for rice blast disease (disease) and rice planthopper (pest) detection. It can be clearly observed that the detection performance of YOLOv-MobileNet V3 is relatively poor, while MobileNet-CA-YOLO exhibits the best detection performance. In the rice blast disease test, MobileNet-CA-YOLO used six detection boxes

and recognized almost all the rice blast disease targets in the image; in the rice planthopper test, MobileNet-CA-YOLO used sixteen detection boxes to identify a large number of visible rice planthopper targets in the image. Overall, MobileNet-CA-YOLO can detect a variety of diseases and pests quickly, accurately, and comprehensively, providing strong technical support for crop protection.

## 4. Discussion

Although the proposed model for rice disease and pest detection in this study has achieved good results in terms of accuracy and lightweight design, there are still some key issues that deserve further discussion and research.

Firstly, the dataset plays a crucial role in the accuracy of the model. Despite using images collected from various online resources and annotated using Makesense AI, there may still be biases and incompleteness in the dataset. Failure to cover all potential disease and pest samples can result in a performance decrease in the model in unknown scenarios. Additionally, manual annotation of the dataset introduces subjectivity, such as emotions, fatigue, or personal biases, which can lead to misclassification or incorrect labeling. In future research, it is important to consider using computer algorithms and machine learning techniques to assist in human classification and annotation, improving accuracy and consistency. For example, combining image recognition algorithms or semi-supervised learning algorithms can yield more reliable results by integrating algorithmic outputs with human classifications and annotations.

Secondly, computational efficiency and memory consumption are important considerations for deploying the model on mobile devices. Although replacing YOLOv7 with MobileNetV3 as the backbone network reduced the number of model parameters, there is still room to explore more lightweight model structures and algorithms to further improve computational efficiency while maintaining high accuracy. Techniques such as model quantization and pruning can be employed to further reduce the model's size and inference time. Additionally, adopting knowledge distillation by designing a complex model as the teacher model and a lightweight model as the student model can ensure that the lightweight model achieves higher performance.

Lastly, while the experiments employed data augmentation techniques to enhance the model's robustness by expanding the dataset, these simple methods may not adequately address complex environmental variations such as occlusions, shadows, and poor lighting conditions. Therefore, further research is warranted to improve the model's adaptability to diverse environments and scenarios, ultimately enhancing the feasibility and reliability of practical applications.

In our future work, we plan to apply the developed digital farmland display system, created in the laboratory, to rice fields and deploy sensor devices in real-world environments. This will enable the collection of more authentic rice field image data and facilitate the expansion of the dataset to enhance its quality. Additionally, we will actively consider aspects, such as crop recognition and weed identification under field conditions, with a particular emphasis on incorporating images of healthy crops. These supplementary research efforts will contribute to improving the reliability and adaptability of the model in real-world applications. By integrating the digital farmland display system into actual rice fields and incorporating sensor devices, we will obtain more accurate and comprehensive data, while simultaneously refining the algorithm for effective practical implementation. These endeavors will enhance the model's versatility and accuracy in adapting to diverse scenarios.

## 5. Conclusions

(1) A dataset for rice disease and insect pest detection was created by collecting images from various online sources, such as Baidu Image Search and Zhihu, and labeling the images using the free data labeling application, Makesense AI. To improve the robustness of the model and the efficiency of detection in complex environments,

image augmentation was applied during dataset creation. The final dataset included three types of insect pests (rice borer, rice planthopper, and rice locust) and three types of diseases (rice blast, rice false smut, and bacterial leaf blight).

(2)    MobileNetV3 was used to replace the backbone network of the original YOLOv7 algorithm. YOLOv7 provides an efficient object detection algorithm, while MobileNetV3 has high computation speed and low memory consumption, resulting in a significant reduction in the number of model parameters. Combining YOLOv7 with MobileNetV3 makes it possible to deploy the rice disease and insect pest detection model on mobile devices.

(3)    The coordinate attention (CA) module was added to the feature fusion layer of YOLOv7 to provide more semantic information for the detection model, thereby improving the accuracy and efficiency of detection. The CIoU loss function was replaced with SIoU to enhance the precision and robustness of the detection model while avoiding overfitting. Ultimately, MobileNet-CA-YOLO achieved good results in testing.

Overall, the proposed rice disease and insect pest detection model is effective, efficient, and suitable for mobile deployment, providing strong technical support for crop protection.

**Author Contributions:** Conceptualization, L.J. and Y.C.; methodology, L.J., H.S. and Y.C.; validation, T.W., X.L. and Y.Z.; formal analysis Y.Z.; investigation, Y.C.; resources, H.S.; writing—original draft preparation, T.W.; writing—review and editing, T.W.; visualization, L.J.; supervision, L.J.; project administration, L.G.; funding acquisition, L.G. All authors have read and agreed to the published version of the manuscript.

**Funding:** This research was funded by the Huzhou Science and Technology Program Public Welfare Projects (2021GZ30, 2021GZ23), the Natural Science Foundation of Zhejiang Province Public Welfare Project (TGN23C130011), and the National Natural Science Foundation of China (31701512).

**Institutional Review Board Statement:** Not applicable.

**Data Availability Statement:** The data presented in this study are available upon request from the corresponding author.

**Acknowledgments:** The authors would like to acknowledge the valuable comments by the editors and reviewers, which have greatly improved the quality of this work.

**Conflicts of Interest:** The authors declare no conflict of interest.

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
