# Peer review of "MobileNet-CA-YOLO: An Improved YOLOv7 Based on the MobileNetV3 and Attention Mechanism for Rice Pests and Diseases Detection"

_agriculture, doi:10.3390/agriculture13071285_

Round 1

Reviewer 1 Report

In this paper, the authors developed a novel rice pest identification model based on the improved YOLOv7 algorithm to obtain timely and accurate rice pest detection results. This study may provide a new solution for rice disease detection, but it has not been validated in a real scenario, and I believe that some work should be added to ensure the completeness of this study. I have listed my concerns below:

1.      The improved model here mainly focuses on reducing the model's size while maintaining accuracy. It is mentioned in the paper that the model is suitable for deployment on mobile devices, but the related work is not presented. Additionally, the open-source lightweight model of YOLOv7 has fewer parameters. It would be helpful to compare its performance against the proposed model and present the results.

2.      Figures and tables: Pay attention to typographical specifications.

3.      Dataset: The images used in the dataset are obtained from internet sources, but the details of the acquisition equipment and shooting methods are unknown. It is essential to consider the requirements for acquisition equipment and images in future practical applications. Can images with the same quality be captured using mobile devices to achieve similar detection results?

4.      Line 17 and Line 271: "[email protected]" and "[email protected]"; can be written uniformly as [email protected] or [email protected].

5.      Line 77: "making it better suited for use with mobile devices"; the term "better" implies a comparison. However, there is no explicit comparison in the text regarding the use of mobile devices.

6.      Figure 1 and Line114: As mentioned earlier, it is difficult to obtain high quality images in realistic field environments where rice pest backgrounds are complex and affected by weather conditions, but the images shown here are of high quality and have been manually filtered to obtain high quality data. Is the dataset thus obtained competent for real scenarios?

7.      Table4: Which YOLOv7 volume does the reference number 141.996M correspond to here?

8.      Line359-360: The accuracy rate of rice fly is 78.2% and the detection rate of rice blast is 86.9%. Are these accuracy rates sufficient for practical applications, and are there any plans for further improvement?

9.      Line393: Is the 91% accuracy rate here the average accuracy rate? The accuracy of detection varies too much between different classes, and this statement might overstate the model’s effectiveness.

10.  Figure12: The image is not clear enough, and what threshold value was set for the detection?

11.  Line403-412: YOLOv7 achieved the highest accuracy on the validation set. Why does it perform much worse on the test set compared to the improved model? Is this an isolated phenomenon or does it reflect the overall performance?

12.  The descriptions in Line 361-363 and Line 369-371 are repetitive.

13.  It would be preferable to make the dataset in the article publicly available.

 Minor editing of English language required.

Author Response

Dear Reviewer,

Thank you for taking the time to review my manuscript. I greatly appreciate your valuable feedback and comments. In response to your questions and suggestions, I have provided detailed explanations and revisions in the attached document.

I have carefully addressed each of the points you raised and made the necessary revisions to enhance the clarity and quality of the manuscript. Additionally, I have included a new discussion section in the paper to comprehensively explore relevant issues.

I hope that the revised manuscript adequately addresses your concerns and meets the requirements of the journal. Please refer to the attached document for a detailed response to your questions and the newly added discussion section. If you have any further inquiries or require additional information, please feel free to contact me.

Thank you once again for your time and consideration. I look forward to hearing your thoughts on the revised manuscript.

Please see the attachment for the detailed responses to your questions.

Sincerely,
Liangquan Jia

Reviewer 2 Report

The authors adopted the well-known MobileNetV3 algorithm to support the chroning process and rice production. The work proposes the MobileNet-Yolo machine learning algorithm as an instrument for identifying pests and diseases occurring during rice cultivation. The model uses the MobileNetV3 convolutional neural network to perform the extraction process of representative graphical features encoded in the form of digital images. The proposed procedure (dedicated to mobile devices) was tested on an empirically obtained (from Internet sources) set of 3773 digital images of rice pests and diseases. The final data file included 3 types of insect pests: rice borer, rice leafhopper and rice locust and 3 types of diseases: rice blight, rice blight and bacterial leaf blight. The proposed model for detecting rice diseases and insect pests has a strong utilitarian aspect: it turns out to be effective and suitable for mobile use, while also providing significant technical support in the process of protecting rice crops.

There are minor stylistic and punctuation errors in the work:

- line 111: Figure 1:[space]Image....[period at the end of the sentence]

- line 157: Figure 2:[space]Image....[period at the end of the sentence]

- line 402: Recognition....

- complete the description of the quantities used in the formulas, e.g.: (1), (2), (3)....

Author Response

Dear Reviewer,

Thank you for taking the time to review my manuscript. I greatly appreciate your valuable feedback and comments. In response to your questions and suggestions, I have provided detailed explanations and revisions in the attached document.

I have carefully addressed each of the points you raised and made the necessary revisions to enhance the clarity and quality of the manuscript. 

I hope that the revised manuscript adequately addresses your concerns and meets the requirements of the journal. Please refer to the attached document for a detailed response to your questions and the newly added discussion section. If you have any further inquiries or require additional information, please feel free to contact me.

Thank you once again for your time and consideration. I look forward to hearing your thoughts on the revised manuscript.

Please see the attachment for the detailed responses to your questions.

Sincerely,
Liangquan Jia

Reviewer 3 Report

MAIN COMMENTS

The topic of the paper is quite interesting and original. 

Please add a table to classify the used dataset of images. (how many images in total?)

Classification and labelling of images was carried out on the basis of a classification made by someone by naked eye: such approach is uncertain, and might produce false positive or false negative. Please discuss this in the paper. 

Also in natural environment there are often obstructions, shadowing effects, excessively high or low lights,... Authors should discuss the presence (or the absence) of this kind of issues in the used images. 

Used images are characterized by high resolution and by a close distance from the target (insect or signs on plant leaves). However normally it happens that an autonomous vehicle randomly captures pictures thata re than analysed. Authors here must explain why they believe that the used images are sufficient to get to some conclusions representative of field image collection. 

It is not clear why the authors adopted Yolo7, and if they have results to compare the eprforamance with previous Yolo versions. 

Also it is not clear the management/recognition/detection of weeds. 

Please in patricular discuss the presence of healthy images

The augmentation process is not somuch significant: rotation, noising, mirroring, etc. are very simple, and probably not really useful to increase the accuracy of the algorithms. 

What about images collected at early or late stages?

What about early detections?

In generealauthors should discuss in the papere the timeliness of images and approaches. 

MINOR COMMENTS

Figure 9 is unreadable

The legends or other fonts of the figures should be readable (while in many cases they are too small)

English needs only minor revision

Author Response

(The authors gave the same response as above.)

Round 2

Reviewer 3 Report

Authors write that they added noise and blur to the dataset. This is not useful to simulate bad environmental conditions (as obstructions, shadowing, bad lighting,...). Noise is a statistical phenomenon; blurring is just the addition of a filter: both of them are easily recognizable: nothing to do with real environement. SO noise and blurring AREE NOT A SOLUTION TO SIMULATE NATURAL ENVIRONMENT. Please correct adding, adding reasonable and pertinent comments  

English is not bad

Author Response

Dear Reviewer,

First and foremost, I would like to express my sincere gratitude for your valuable comments and suggestions during the first round of review of our paper. Your feedback has been instrumental in improving the quality and academic standard of our research work.

During the second round of review, we carefully reviewed and considered your comments. Attached, you will find a detailed response addressing each of the points you raised. Once again, thank you for providing us with valuable insights and suggestions during the review process. Your expertise and rigorous approach have had a profound impact on our research work, and we are sincerely grateful for your guidance.

We have thoroughly analyzed and discussed the issues you raised, and our understanding and proposed solutions are outlined in the attachment. We greatly value your professional knowledge and meticulousness and appreciate your guidance. Your review comments have significantly influenced our research work, helping us further refine and enhance the quality of the paper.

Once again, we would like to express our sincere appreciation for your efforts and valuable input during the review process. We will incorporate your suggestions and ensure that the paper meets academic standards in all aspects. Should you have any further questions or require additional discussion, we would be more than willing to engage in further dialogue.

Thank you wholeheartedly for your support and guidance!

Point 1: Authors write that they added noise and blur to the dataset. This is not useful to simulate bad environmental conditions (as obstructions, shadowing, bad lighting,...). Noise is a statistical phenomenon; blurring is just the addition of a filter: both of them are easily recognizable: nothing to do with real environement. SO noise and blurring AREE NOT A SOLUTION TO SIMULATE NATURAL ENVIRONMENT. Please correct adding, adding reasonable and pertinent comments.

Response 1:

Dear Reviewer, thank you very much for your feedback on our manuscript. We understand that various methods are commonly used to address occlusion in object detection, such as data augmentation-based algorithms, holistic feature-based algorithms, and improvement algorithms based on target structures. Data augmentation techniques, such as adding black pixels or random noise, can enhance the robustness of the models. For instance, a method called "Cutout" has been proposed, which effectively tackles overfitting in convolutional neural networks and the problem of insensitivity to occlusion by randomly masking regions in images. Holistic feature-based detection algorithms can be categorized into three directions: improvement based on target structures, improvement based on loss functions, and improvement based on non-maximum suppression algorithms. Improvement based on target structures involves designing detectors that leverage prior knowledge and target visible structure information to enhance occlusion detection performance.

Regarding shadows and lighting issues, common techniques include histogram equalization, gamma correction algorithms, and the Retinex algorithm.

In our current research work, our focus has mainly been on model lightweighting and recognition accuracy, and we have not yet utilized more sophisticated and complex image processing methods for dataset preprocessing. We have only employed simple methods such as adding noise and color jittering. However, we acknowledge that the best approach is to acquire more effective image data using sensor facilities, and this will be a key focus of our future work. We will carefully consider your suggestions and pay more attention to these issues in our future research.

Lastly, we greatly appreciate the valuable feedback provided by the reviewer. Your suggestions have provided important guidance for our work, and we will strive to improve our methods to address challenges in natural environments, including occlusion, shadows, and lighting. If you have any further suggestions or comments, we would be more than happy to hear them.

The content we have modified is as follows:

Line 154 – 157: Content transformation includes color jittering, adding noise, and so on. The combination of geometric transformations and content transformations allows for a comprehensive modification of image attributes, enhancing the robustness and generalization capability of the model.

Line 448-473: Lastly, while the experiments employed data augmentation techniques to enhance the model's robustness by expanding the dataset, these simple methods may not adequately address complex environmental variations such as occlusions, shadows, and poor lighting conditions. Therefore, further research is warranted to improve the model's adaptability to diverse environments and scenarios, ultimately enhancing the feasibility and reliability of practical applications.
